# Multimorbidity and Serological Response to SARS-CoV-2 Nine Months after 1st Vaccine Dose: European Cohort of Healthcare Workers—Orchestra Project

**DOI:** 10.3390/vaccines11081340

**Published:** 2023-08-08

**Authors:** Concepción Violán, Lucía A. Carrasco-Ribelles, Giulia Collatuzzo, Giorgia Ditano, Mahsa Abedini, Christian Janke, Christina Reinkemeyer, Le Thi Thu Giang, Filippo Liviero, Maria Luisa Scapellato, Marcella Mauro, Francesca Rui, Stefano Porru, Gianluca Spiteri, Maria Grazia Lourdes Monaco, Angela Carta, Marina Otelea, Agripina Rascu, Eleonóra Fabiánová, Zuzana Klöslová, Paolo Boffetta, Pere Torán-Monserrat

**Affiliations:** 1Unitat de Suport a la Recerca Metropolitana Nord, Institut Universitari d’Investigació en Atenció Primària Jordi Gol (IDIAP Jordi Gol), Mare de Déu de Guadalupe, 08303 Mataró, Spain; lcarrasco@idiapjgol.info (L.A.C.-R.); ptoran.bnm.ics@gencat.cat (P.T.-M.); 2Germans Trias i Pujol Research Institute (IGTP), Camí de les Escoles, s/n, 08916 Badalona, Spain; 3Grup de REcerca en Impacte de les Malalties Cròniques i les Seves Trajectòries (GRIMTra) (2021 SGR 01537), Institut Universitari d’Investigació en Atenció Primària Jordi Gol (IDIAP Jordi Gol), Mare de Déu de Guadalupe, 08303 Barcelona, Spain; 4Network for Research on Chronicity, Primary Care, and Health Promotion (RICAPPS) (RD21/0016/0029), Insitituto de Salud Carlos III, Av. de Monforte de Lemos, 5, 28029 Madrid, Spain; 5Direcció d’Atenció Primària Metropolitana Nord Institut Català de Salut, Ctra. de Barcelona, 473, Sabadell, 08204 Barcelona, Spain; 6Universitat Autónoma de Barcelona, Plaça Cívica, 08193 Bellaterra, Spain; 7Department of Medical and Surgical Sciences, University of Bologna, 40138 Bologna, Italy; giulia.collatuzzo@studio.unibo.it (G.C.); giorgia.ditano2@unibo.it (G.D.); mahsa.abedini@unibo.it (M.A.); paolo.boffetta@unibo.it (P.B.); 8Division of Infectious Diseases and Tropical Medicine, LMU Klinikum, Leopoldstraße 5, 80802 Munich, Germany; christian.janke@lrz.uni-muenchen.de (C.J.); christina.reinkemeyer@med.uni-muenchen.de (C.R.); 9Department of Pediatrics, Dr. von Hauner Children’s Hospital, University Hospital, LMU Munich, Lindwurmstrasse 4, 80337 Munich, Germany; tlethi@med.lmu.de; 10Department of Cardiac Thoracic Vascular Sciences and Public Health, University of Padova, 35128 Padova, Italy; filippo.liviero@unipd.it; 11Occupational Medicine Unit, University Hospital of Padova, 35128 Padova, Italy; marialuisa.scapellato@unipd.it; 12Unit of Occupational Medicine, Department of Medical Sciences, University of Trieste, 34129 Trieste, Italy; mmauro@units.it (M.M.); frui@units.it (F.R.); 13Occupational Medicine Unit, University Hospital of Verona, 37134 Verona, Italy; stefano.porru@univr.it (S.P.); gianluca.spiteri@aovr.veneto.it (G.S.); mariagrazialourdes.monaco@aovr.veneto.it (M.G.L.M.); angela.carta@univr.it (A.C.); 14Section of Occupational Health, Department of Diagnostics and Public Health, University of Verona, 37134 Verona, Italy; 15University of Medicine and Pharmacy “Carol Davila”, 020022 Bucharest, Romania; marina.otelea@umfcd.ro (M.O.); agrirascu@yahoo.com (A.R.); 16Occupational Health Department, Regional Authority of Public Health, 97556 Banská Bystrica, Slovakia; fabianova@vzbb.sk (E.F.); kloslova@vzbb.sk (Z.K.); 17Public Health Department, Faculty of Health, Catholic University, 03401 Ružomberok, Slovakia; 18Department of Medicine, Faculty of Medicine, Universitat de Girona, 17001 Girona, Spain; 19Multidisciplinary Research Group in Health and Society (GREMSAS) (2021 SGR 01484), Institut Universitari d’Investigació en Atenció Primària Jordi Gol (IDIAP Jordi Gol), Mare de Déu de Guadalupe, 08303 Barcelona, Spain

**Keywords:** SARS-CoV-2, antibodies, IgG, humoral immunity, seroprevalence, COVID-19, vaccine, multimorbidity, healthcare workers, cohort

## Abstract

Understanding antibody persistence concerning multimorbidity is crucial for vaccination policies. Our goal is to assess the link between multimorbidity and serological response to SARS-CoV-2 nine months post-first vaccine. We analyzed Healthcare Workers (HCWs) from three cohorts from Italy, and one each from Germany, Romania, Slovakia, and Spain. Seven groups of chronic diseases were analyzed. We included 2941 HCWs (78.5% female, 73.4% ≥ 40 years old). Multimorbidity was present in 6.9% of HCWs. The prevalence of each chronic condition ranged between 1.9% (cancer) to 10.3% (allergies). Two regression models were fitted, one considering the chronic conditions groups and the other considering whether HCWs had diseases from ≥2 groups. Multimorbidity was present in 6.9% of HCWs, and higher 9-months post-vaccine anti-S levels were significantly associated with having received three doses of the vaccine (RR = 2.45, CI = 1.92–3.13) and with having a prior COVID-19 infection (RR = 2.30, CI = 2.15–2.46). Conversely, lower levels were associated with higher age (RR = 0.94, CI = 0.91–0.96), more time since the last vaccine dose (RR = 0.95, CI = 0.94–0.96), and multimorbidity (RR = 0.89, CI = 0.80–1.00). Hypertension is significantly associated with lower anti-S levels (RR = 0.87, CI = 0.80–0.95). The serological response to vaccines is more inadequate in individuals with multimorbidity.

## 1. Introduction

Vaccination against Severe acute respiratory syndrome coronavirus 2 (SARS-CoV-2) has been one of the milestones in the history of infectious disease prevention. Both the rapid development of vaccines and the speed with which vaccines have been administered have made this possible. Currently, the ω variant is dominant in the world. The disease caused by this variant is less severe and results in fewer hospitalizations and deaths, but it is more transmissible [1,2,3,4,5].

Healthcare workers (HCWs) are a professional group particularly exposed to SARS-CoV-2 infection [6]. Therefore, they were one of the first groups to be administered the first dose of the vaccine and the booster doses [7]. The European Centre for Disease Prevention and Control (ECDC) and the European Medicines Agency (EMA) have issued a joint statement regarding adapted COVID-19 vaccines and considerations for their use during the upcoming autumn 2023 vaccination campaigns, in which healthcare workers are a priority vaccination group [8].

Vaccines against SARS-CoV-2 are effective in preventing severe cases of COVID-19 and reducing mortality [9,10,11]. The persistence of specific antibodies is considered a marker of the immune system’s ability to protect against a particular microorganism [12,13,14,15,16,17].

The long-term immunity induced by vaccines is essential for protection against rapidly emerging mutant strains of SARS-CoV-2. In a previous study conducted in the ORCHESTRA cohort, we demonstrated that, at 9 months, antibody levels showed a decreasing trend over time without analyzing whether this was dependent on previous diseases or multimorbidity [18]. Several studies have investigated the humoral response to the SARS-CoV-2 vaccine in patients suffering from cancer, kidney failure, and immunosuppression, showing a lower immune response in these subjects [19,20,21]. Similar results were reported among subjects who underwent kidney transplant or who were affected by multiple sclerosis [19,20,21,22]. It is worth noting that there are likely many other factors that can influence the immune response to the virus and the vaccines, and more research is needed to fully understand this complex topic. However, only one study analyzed the immune response to SARS-CoV-2 in people with multimorbidity [23].

Multimorbidity, which refers to the presence of two or more chronic conditions in an individual [24], has been linked to a greater risk of severe SARS-CoV-2 infection [25,26]. A study conducted in the UK found that multimorbidity was independently associated with a greater risk of severe SARS-CoV-2 infection [27]. The study also found that the risk remained consistent across potential effect modifiers, except for a greater risk among older age groups. It is important to note that, while comorbidities are associated with severe SARS-CoV-2 infection, not all individuals with these will necessarily experience severe illness. However, a study conducted in Northern Italy found that patients with multimorbidity were less likely to be infected with SARS-CoV-2, possibly due to greater attention to protective methods [28].

A previous study has demonstrated that multimorbidity is associated with weaker initial antibody response and cellular immunity, which could impede virus clearance and lead to a more severe disease progression [29]. Other authors have analyzed the impact of comorbidities on humoral antibody responses against the receptor-binding domain (RBD) of SARS-CoV-2, concluding that multimorbidity has a negative impact on antibody response [23]. In another study, it was identified that individuals with diabetes mellitus and kidney disease exhibited lower humoral response against the RBD of SARS-CoV-2 [28]. These previous observations regarding multimorbidity and immune response suggest the hypothesis that individuals with multiple chronic conditions may have a diminished immune response following vaccination.

Little is known about the changes on humoral response after vaccination, depending on the type of previous disease, and even less is known about how the presence of two or more chronic diseases affects the humoral response. Therefore, understanding whether the humoral response is affected by previous diseases or not is important for future vaccination policies against SARS-CoV-2, as it helps to identify a group of individuals who are more prone to worse outcomes when infected with SARS-CoV-2.

Thus, we aim to identify the association of multimorbidity with serological response to SARS-CoV-2 nine months after first vaccine dose in a large European cohort.

## 2. Materials and Methods

### 2.1. Study Design and Ethics

ORCHESTRA comprises a prospective multicenter cohort including more than 80,000 HCWs employed in hospitals, and Primary Care Health Centers in different European countries [30]. This analysis involves HCWs from one center in Germany (Munich), three centers in Italy (Padova, Trieste, and Verona), several centers in Slovakia and Romania (the two latter treated as individual cohorts), and one center from Spain (Northern Barcelona region) with serological results at 9 months after first vaccination dose. The pooled study was approved by the Italian Medicine Agency (AIFA) and the Ethics Committee of the Italian National Institute of Infectious Diseases (INMI) Lazzaro Spallanzani. Each cohort was approved by the local ethical boards. All participants recruited in the study were fully informed about the ORCHESTRA protocol and signed informed consent to participate. They consented to use their collected data for research and agreed to the applicable regulations, privacy policies, and terms of use. Participant data was anonymized before sharing for the analysis and stored in a database securely.

### 2.2. Participant Recruitment, Follow-Up

Health professionals (physicians, nurses, nursing assistants, researchers, and other essential workers) in direct contact with patients during the first, second, or third wave of COVID-19 were recruited between 27 December 2020 and 31 May 2022. Inclusion criteria included available and positive serology results during a 9-month timeframe from first dose administration, defined as an interval of 210–330 days, and available information about chronic diseases and conditions.

Only individuals who had antibodies to SARS-CoV-2 were included in the analyses presented in this paper. SARS-CoV-2 antibody levels assays were different between centers (Table 1). Only individuals who had a positive level of antibodies to SARS-CoV-2 according to the validated thresholds defined by the manufacturer for each assay were included in the analysis. This approach has been followed in previous works from the ORCHESTRA project [18,31]

A database was created for the ORCHESTRA cohort. (1) Data on sociodemographic characteristics, results of PCR testing, and vaccination status, including date of vaccination doses and type, were either abstracted from electronic health records or collected using self-administered questionnaires or ongoing loco-regional databases. (2) Results on the level of anti-S antibodies were either collected from medical records or generated through ad hoc testing. (3) We also implemented a study-specific Redcap^®^ data collection system, which includes the patient’s unique numerical identifier and the results of serological and immunological tests. The following variables were collected in this database: (a) vaccines (brand name, dates, doses); (b) SARS-CoV-2 infection, considering PCR results and anti-N serologies; (c) chronic conditions, in the following groups: cardiovascular diseases, respiratory diseases, hypertension, diabetes, diseases involving the immune system, cancer, and allergies (allergies only available in Germany, Spain, Romania, and Slovakia) collected at the baseline through self-reported questionnaires or electronic health records according to each country’s definition, (d) sex, (e) age, in 4 different categories (in 10-year age groups: <29, 30–39, 40–49, and ≥50) and 10-year increases.

All cohorts included in the ORCHESTRA project have undergone extensive data harmonization [18].

### 2.3. Statistical Analysis

Absolute and relative frequencies were used to describe the cohorts. Anti-S antibody levels underwent a standardization process: first, they were log-transformed to account for the skewness of the distribution; secondly, to consider the heterogeneity in measurement methods, log-transformed results were normalized by dividing them by the center-specific standard errors. This standardization was also used in previous analyses within the ORCHESTRA project [18] to obtain comparable measures among the cohorts even if different tests were used.

We fitted two multivariate linear regression models to estimate the relative risks (RR) and corresponding 95% confidence intervals (CI) of an increase of one standard deviation (SD) of normalized log-transformed antibody level 9-month post-vaccination. The first model considered the number of (grouped) chronic conditions registered for each HCWs, grouped as 0 or 1, or ≥2. The second considered each chronic condition group as binary variables. The following confounders were considered in each model: sex, age (10-year increases), time since the last vaccine dose, previous COVID-19 infection (detected by PCR or anti-N serology test), and number of vaccine doses received. All tests were two-sided, and a statistical probability of *p* < 0.05 was considered significant. All analyses were performed using Stata^®^ software V. 17 (StataCorp LP, College Station, TX, USA).

## 3. Results

### 3.1. Description of the Participants

The analysis included 2941 vaccinated HCWs from seven European cohorts included in the ORCHESTRA project. Among these, 78.5% were female, 73.4% were aged ≥40 years old, and 77.5% had no prior COVID-19 infection. Infection rates varied among cohorts, from 5.7% (Germany—Munich) to 50.7% (Spain—Northern Barcelona). Regarding vaccination, 98.3% had received two or more doses. Multimorbidity (≥2 chronic conditions groups) was present in 6.9% of HCWs, although its prevalence varied greatly between cohorts: from 1.7% (Italy—Trieste) to 27.4% (Spain—Northern Barcelona). The prevalence of each chronic condition ranged between 1.9% (cancer) and 10.3% (allergies). The description of each cohort can be found in Table 2. Moreover, as Table 2 shows, between 95% and 100% of individuals in each cohort, and 98.4% of the total population considered, were vaccinated with mRNA-based vaccines. Table A1 shows the 10 most common combinations of chronic conditions, with allergies or hypertension being present in the top five pairs.

Mean standardized anti-S levels at 9 months after first dose was 4.67, ranging from 2.39 to 10.78 between cohorts (see Table 1). On average, 257 and 175 days had passed from the first and last vaccine dose, respectively, up to the 9-month serology measurement.

### 3.2. Association of Chronic Conditions with 9-Month Post-Vaccine Anti-S Levels

As Table 3 shows, higher levels of anti-S after 9 months of vaccination were significantly associated with having received three doses of SARS-CoV-2 vaccine, any type (RR = 2.45, CI = 1.92–3.13, *p*-value < 0.0001), and having a prior COVID-19 infection (RR = 2.30, CI = 2.15–2.46, *p*-value < 0.0001). Conversely, anti-S levels significantly decreased with each 10-year increase in age (RR = 0.94, CI = 0.91–0.96, *p*-value < 0.0001), more time since the last vaccine dose had passed (RR = 0.95, CI = 0.94–0.96, *p*-value < 0.0001), and when the HCWs had diseases on ≥2 chronic conditions groups (RR = 0.89, CI = 0.80–1, *p*-value = 0.043). Among the groups of chronic conditions considered, only hypertension showed a significant association with lower anti-S levels (RR = 0.87, CI = 0.80–0.95, *p*-value = 0.002). Sex did not have a significant impact on the anti-S levels 9 months post-vaccination.

## 4. Discussion

This pooled analysis included the results of COVID-19 anti-S antibodies at 9 months after the first vaccine dose in 2941 HCWs from seven different European cohorts included in the ORCHESTRA project.

We observed that a higher serological response was associated with previous COVID-19 infection and the administration of the third vaccine dose. Additionally, older age and the time elapsed since the first dose were inversely associated with the antibody level. These results are consistent with previous studies conducted in the ORCHESTRA project, albeit these previous analyses included a large number of HCWs than the current study population [18]. Seven groups of chronic diseases were identified at baseline. Having two or more chronic diseases was associated with a diminished serological response. Moreover, hypertension was the only disease that demonstrated a lower serological response at 9 months post-vaccination.

The progressive decline of vaccine immunity is common and widely acknowledged. Among the factors that influence the immune response to vaccination, which varies significantly among individuals, are intrinsic host factors (such as age, sex, genetics, and comorbidities), perinatal factors (such as birth weight, feeding method, and maternal factors), and extrinsic factors (such as pre-existing immunity, microbiota, and antibiotics) [23,32] There is limited literature regarding the serological response to SARS-CoV-2 in individuals with multimorbidity. A preprint publication conducted in Bangladesh focused on individuals aged ≥ 18 years, with a cohort of 1005 participants, of which only 72 had multimorbidity [23]. The study investigated the impact of comorbidities on humoral antibody response against the specific receptor-binding domain (RBD) of SARS-CoV2. Diabetes mellitus and kidney disease had a significant negative impact on the decline of humoral SARS-CoV-2-specific IgG and total antibody (TAb) response. IgG and TAb declined more rapidly in diabetic and kidney disease patients compared to the other morbidity groups. Follow-up serological measurements demonstrated that antibody response declined within 4 months after receiving the second dose. Although this study was not peer-reviewed and is descriptive, it indicates the same trend of a decreased serological response in individuals with multimorbidity [23].

Several studies show that hypertension is associated with a lower antibody response following vaccination with SARS-CoV-2 [32,33] Individuals with hypertension may also be at higher risk for breakthrough infections following vaccination. Nevertheless, in these studies, only cross-sectional determinations have been made and the kinetics have not been measured to identify the moment in which the decrease in antibodies begins. Periodic monitoring of the antibody levels might be a good indicator to guide personalized needs for further booster shots to maintain adaptive immunity. Nonetheless, it is important that people obtain their COVID-19 vaccination, especially people with hypertension. However, more research is needed to fully understand the relationship between hypertension and the serological response to SARS-CoV-2 vaccination.

In the case of diabetes, we did not observe a decrease in the serological response in this study, probably due to the small number of participants with diabetes and the age of the study population. However, other studies conducted on individuals over 65 years old have found that diabetes affects the antibody response to SARS-CoV-2, particularly in non-insulinized individuals [34,35].

Regarding cancer, our study has few patients, and we found no differences in serological response. However, some studies report cancer patients may have a lower serological response to SARS-CoV-2 vaccination, especially those with hematologic malignancies [36,37].

Regarding individuals with allergies, no increased antibody response was observed after the administration of the SARS-CoV-2 vaccine. However, a study conducted in Spain, also among HCWs, revealed that allergic individuals exhibited a higher serological response following infection [38]. This could be justified by their heightened susceptibility to secondary COVID-19 due to chronic inflammation of the respiratory pathways resulting from continuous exposure to allergens [39] Nonetheless, this pathophysiological mechanism should not occur after vaccine administration, and thus, antibody levels should not be higher.

For the other disease groups analyzed—cardiovascular diseases, respiratory diseases, and diseases involving the immune system—we have not found comparable studies, namely, articles that considered these grouped diseases and their relationship with the serological responses to vaccines. Thus, it is important to note that all studies that analyze serological response and diseases have different patient populations, methodologies, and low statistical power for these conditions, so the findings may not be directly comparable.

The relationship between multimorbidity and COVID-19 was identified in early studies. However, establishing causality, determining underlying mechanisms, and understanding the clinical implications are more complex due to the multitude of confounding factors (age, sex, smoking, and medication) and the individual characteristics of patients that contribute to significant variability. Several physio-pathological mechanisms have been described that affect different phases of the disease and its sequelae, known as post-COVID syndrome. On 5 May 2023, the WHO declared that the health emergency had ended; however, the threat to public health continues. Therefore, it is now necessary to continue conducting studies that take into account host genetics as a pathway for causal inference by eliminating many sources of confounding. The ongoing investigation of the disease–host interactions and disease states will be crucial in informing the stratification of therapeutic approaches and improving outcomes for patients [29].

The investigation of the serological response to SARS-CoV-2 vaccination in individuals with multimorbidity is a subject of scientific exploration. Current research aims to elucidate how multimorbidity may impact the immune response triggered by the SARS-CoV-2 vaccine. Several studies indicate that individuals with underlying health conditions or multiple chronic illnesses may exhibit a diminished immune response to vaccination compared to those without such conditions. Consequently, this could potentially affect the generation and longevity of vaccine-induced antibodies [29].

Analyzing these associations would help to understand the determinants influencing the serological response in individuals with specific diseases and multimorbidity, apart from identifying the most advantageous conditions for these individuals before, during, and after vaccination to achieve an optimal antibody response. Many unresolved issues regarding potential factors affecting vaccine immunogenicity, including vaccine type, number of doses administered, revaccination intervals, and hyperglycemia in patients with diabetes, should be addressed in future studies research. Moreover, we would like to highlight that the study population consists of healthcare workers, which might be healthier than the general population Further investigations are necessary to achieve a more comprehensive understanding of how multimorbidity influences the serological response to SARS-CoV-2 vaccination.

The present study has several notable strengths. Firstly, it employed a prospective design, allowing for the collection of data over time, which enhances the validity of the findings. Additionally, the study benefited from a large sample size, achieved by combining data from seven cohorts of healthcare workers from four European countries. This large sample size increases the statistical power and generalizability of the study’s results.

To address the variability in blood analysis methods among the cohorts, a crucial step was taken to standardize the measurement of antibody levels. This standardization ensured that the results obtained were comparable across the cohorts, enabling the identification of genuine differences in antibody levels based on various characteristics and factors predictive of a robust immune response at 9 months post-vaccination. Standardization has been successfully utilized in previous studies and serves as an effective approach for addressing heterogeneity in testing methodologies when comparing studies conducted in diverse populations.

The overall consistency of our results with previous literature supports the robustness of the analysis. Furthermore, internal validity was checked through various sensitivity analyses, made possible by the large number of subjects included in the analysis. Finally, this study is part of a series of periodic updates on serological data from vaccinated healthcare workers in the joint ORCHESTRA study. ORCHESTRA, which has already produced results at 6 and 9 months post-vaccination [18,32], will provide further results as the follow-up of vaccinated healthcare workers continues in the future, including individual-level trends in antibody levels. Furthermore, the motivation of healthcare workers (HCWs) to collect their health data should also be regarded as a strength of the study.

A limitation of the study is the lack of data on multimorbidity for all the cohorts included in the ORCHESTRA project, which would have facilitated a larger study size and, consequently, greater statistical power [31]. Another aspect to consider is the lack of homogeneity in the definition of chronic conditions; however, this is a widespread problem in all multimorbidity studies [40]. Another aspect to consider is the age and health status of the cohort. As it is a cohort of healthcare workers under 65 years, the prevalence of chronic diseases is lower than in other studies involving an elderly population [40]. The linkage of this information with genetic databases will allow for the segmentation and characterization of individuals at higher risk. Additionally, healthcare workers tend to be healthier than the general population [41].

## 5. Conclusions

The results of this study, although limited, support the hypothesis that individuals with chronic diseases such as hypertension may also have a lower serological response to vaccines administered for SARS-CoV-2, often accompanied by other cardiovascular diseases or other prevalent diseases. Consequently, this criterion should be considered in guidelines for revaccination against SARS-CoV-2. However, before making these decisions, it would be necessary to conduct longer-term serological studies as the serological response is influenced by various factors such as age, sex, smoking, and medication. This would need segmentation of the population to propose personalized clinical recommendations.

This information is important for public health policies as it can help identify populations at higher risk of reduced vaccine effectiveness and guide vaccination strategies, such as the need for additional doses or tailored approaches for individuals with multimorbidity.

## Figures and Tables

**Table 1 vaccines-11-01340-t001:** Crude and standardized 9-month post-vaccine anti-S levels, per cohort, and number of days since first and last dose to this serology.

Cohort	Assay	Crude 9-Month Post-Vaccine Anti-S Serology Test *	Standardized 9-Month Post-Vaccine Anti-S Serology Test *	Days Since 1st Dose to 9-Month Serology Test ^†^	Days Since Last Dose to 9-Month Serology Test ^†^
Germany— Munich *n* = 549 (18.7%)	Ro-RBD-Ig- quant-DBS	170.90 (16.44)	2.39 (0.05)	275 (25) [210, 330]	203 (88) [1, 316]
Italy—Padova *n* = 163 (5.5%)	Diasorin Liaison^®^ SARS-CoV-2 Trimeric—S—Igg	8434.65 (951.08)	10.78 (0.13)	302 (28) [217, 330]	57 (37) [17, 302]
Italy—Trieste *n* = 288 (9.8%)	CMIA Abbott anti S-RBD	6236.01 (1881.91)	4.74 (0.08)	258 (22) [210, 330]	202 (75) [3, 293]
Italy—Verona *n* = 1403 (47.7%)	Diasorin Liaison^®^ SARS-CoV-2 Trimeric—S—Igg	2443.59 (135.82)	4.62 (0.03)	258 (22) [210, 328]	202 (75) [3, 293]
Romania— Multicenter *n* = 52 (1.8%)	Abbot SARS-CoV-2 IgG II Quant test	8746.71 (3772.14)	5.06 (0.31)	255 (44) [210, 330]	199 (65) [54, 309]
Slovakia— Multicenter *n* = 11 (0.4%)	Anti-SARS-CoV-2 QuantiVac ELISA (IgG) EUROIMMUN	297.69 (23.34)	7.15 (0.10)	226 (8) [212, 238]	196 (9) [181, 210]
Spain— Northern Barcelona *n* = 475 (16.1%)	DECOV1901 ELISA (IgG-S)	1851.23 (110.71)	5.72 (0.05)	265 (31) [210, 329]	207 (76) [2, 324]
Total *n* = 2941 (100%)	-	-	4.67 (0.04)	257 (38) [210, 330]	175 (81) [1, 324]

* Mean (standard error); ^†^ Mean (standard deviation) [minimum, maximum].

**Table 2 vaccines-11-01340-t002:** Description of the study population, per cohort.

	Germany— Munich *n* = 549 (18.7%)	Italy— Padova *n* = 163 (5.5%)	Italy— Trieste *n* = 288 (9.8%)	Italy— Verona *n* = 1403 (47.7%)	Romania— Multicenter *n* = 52 (1.8%)	Slovakia— Multicenter *n* = 11 (0.4%)	Spain— Northern Barcelona *n* = 475 (16.1%)	Total *n* = 2941 (100%)
Gender (Female)	440 (80.2)	138 (84.7)	203 (70.5)	1118 (79.7)	40 (76.9)	8 (72.7)	362 (76.2)	2309 (78.5)
Age group								
≤29	69 (12.6)	9 (5.5)	15 (5.2)	159 (11.3)	2 (3.8)	1 (9.1)	44 (9.3)	299 (10.2)
30–39	139 (25.3)	25 (15.3)	43 (14.9)	201 (14.3)	4 (7.7)	2 (18.2)	68 (14.3)	482 (16.4)
40–49	120 (21.9)	41 (25.2)	75 (26.0)	361 (25.7)	17 (32.7)	5 (45.4)	170 (35.8)	789 (26.8)
≥50	221 (40.3)	88 (54.0)	155 (53.8)	682 (48.6)	29 (55.8)	3 (27.3)	193 (40.6)	1371 (46.6)
Job title								
Administration	NA	34 (20.9)	13 (4.5)	152 (10.8)	7 (14.5)	1 (9.1)	66 (15.1)	274 (11.6)
Technician	NA	15 (9.2)	55 (19.1)	143 (10.2)	11 (21.1)	1 (9.1)	0 (0.0)	225 (9.5)
Nurse	NA	82 (50.3)	105 (36.5)	597 (42.5)	7 (13.5)	5 (45.4)	186 (42.6)	983 (41.7)
Physician (includes residents)	NA	12 (7.4)	69 (24.0)	263 (18.8)	26 (50.0)	1 (9.1)	142 (32.5)	513 (21.8)
Other HCWs (includes auxiliary workers)	NA	20 (12.3)	46 (16.0)	248 (17.7)	1 (1.9)	3 (27.3)	43 (9.8)	361 (15.3)
Previous COVID-19 infection (PCR)								
Never infected	518 (94.3)	106 (65.0)	235 (81.6)	1162 (82.8)	42 (80.8)	7 (63.6)	209 (44.0)	2279 (77.5)
Infected once	31 (5.7)	55 (33.8)	53 (18.4)	235 (16.8)	10 (19.2)	4 (36.4)	241 (50.7)	629 (21.4)
Infected twice	0 (0.0)	2 (1.2)	0 (0.0)	6 (0.4)	0 (0.0)	0 (0.0)	25 (5.3)	33 (1.1)
Previous COVID-19 infection (positive anti-N serology)								
Never infected	507 (92.3)	NA	NA	NA	NA	NA	328 (69.6)	835 (81.9)
Infected at least once	42 (7.7)	NA	NA	NA	NA	NA	143 (30.4)	185 (18.1)
Number of vaccine doses								
1 dose	1 (0.2)	0 (0.0)	1 (0.3)	15 (1.1)	0 (0.0)	0 (0.0)	32 (6.7)	49 (1.7)
2 doses	429 (78.1)	38 (23.3)	252 (87.5)	1123 (80.0)	43 (82.7)	11 (100.0)	407 (85.7)	2303 (78.3)
3 doses	119 (21.7)	125 (76.7)	35 (12.2)	265 (18.9)	9 (17.3)	0 (0.0)	36 (7.6)	589 (20.0)
Vaccination scheme								
mRNA-based only	528 (96.2)	44 (100)	286 (100)	1315 (100)	51 (98.1)	11 (100)	452 (95.2)	2687 (98.4)
Adenovirus-based only	6 (0.1)	0 (0.0)	0 (0.0)	0 (0.0)	0 (0.0)	0 (0.0)	7 (1.5)	13 (0.4)
Mixed type	15 (0.3)	0 (0.0)	0 (0.0)	0 (0.0)	1 (1.9)	0 (0.0)	16 (3.3)	32 (1.2)
Missing	-	119 (73.0)	2 (0.7)	88 (6.3)	-	-	-	209 (7.1)
Number of comorbidities								
0 or 1	530 (96.5)	157 (96.3)	283 (98.3)	1378 (98.2)	36 (69.2)	10 (90.9)	345 (72.6)	2739 (93.1)
2 or more	19 (3.5)	6 (3.7)	5 (1.7)	25 (1.8)	16 (30.8)	1 (9.1)	130 (27.4)	202 (6.9)
Comorbidities								
Allergies *	11 (2.0)	NA	NA	NA	13 (25.0)	0 (0.0)	88 (18.5)	112 (10.3)
Diabetes	4 (0.7)	4 (2.5)	6 (2.1)	29 (2.1)	3 (5.8)	0 (0.0)	14 (2.9)	30 (2.0)
Cardiovascular diseases	15 (2.7)	5 (3.1)	13 (4.5)	44 (3.1)	11 (21.2)	1 (9.1)	46 (9.7)	135 (4.6)
Hypertension	30 (5.5)	18 (11.2)	22 (7.6)	107 (7.6)	14 (26.9)	3 (27.3)	106 (22.3)	300 (10.2)
Respiratory diseases	18 (3.3)	7 (4.3)	11 (3.8)	52 (3.7)	10 (19.2)	0 (0.0)	65 (13.7)	163 (5.5)
Diseases involving immune system	13 (2.4)	4 (2.5)	7 (2.4)	32 (2.3)	3 (5.8)	0 (0.0)	113 (23.8)	172 (5.8)
Cancer	7 (1.3)	4 (2.5)	4 (1.4)	5 (0.4)	2 (3.8)	0 (0.0)	33 (6.9)	55 (1.9)

NA: Not Available. * = Available in all countries but Italy

**Table 3 vaccines-11-01340-t003:** Multiple linear regression of the standardized 9-month serology test result.

	Multimorbidity as Binary Variable			Multimorbidity as Chronic Conditions		
	Risk Ratio (RR)	*p*-Value	Robust Standard Error	Risk Ratio (RR)	*p*-Value	Robust Standard Error
Cohort * Italy—Verona	Ref.			Ref.		
Germany—Munich	0.13 [0.12, 0.14]	<0.0001	0	0.13 [0.12, 0.14]	<0.0001	0
Italy—Padova	151.33 [134.25, 170.60]	<0.0001	9.25	150.98 [133.82, 170.34]	<0.0001	9.29
Italy—Trieste	1.41 [1.29, 1.54]	<0.0001	0.06	1.40 [1.28, 1.54]	<0.0001	0.06
Romania—Multicenter	1.63 [1.34, 1.98]	<0.0001	0.16	1.64 [1.35, 1.99]	<0.0001	0.16
Slovakia—Multicenter	17.93 [11.89, 27.05]	<0.0001	3.76	18.33 [12.15, 27.65]	<0.0001	3.84
Spain—Barcelona	2.95 [2.71, 3.22]	<0.0001	0.13	2.94 [2.70, 3.21]	<0.0001	0.13
Gender *—*Female*	1.04 [0.98, 1.11]	0.18	0.03	1.03 [0.97, 1.10]	0.31	0.03
Age—*10 years increase*	0.94 [0.91, 0.96]	<0.0001	0.01	0.94 [0.92, 0.97]	<0.0001	0.01
Days since last dose to 9-month serology—*10 days increase*	0.95 [0.94, 0.96]	<0.0001	0	0.95 [0.94, 0.96]	<0.0001	0
Previous COVID-19 Infection *—*Infected at least once: positive PCR/anti-N serology*	2.30 [2.15, 2.46]	<0.0001	0.08	2.32 [2.16, 2.48]	<0.0001	0.08
Number of vaccine doses *						
2 doses	1.12 [0.92, 1.37]	0.26	0.12	1.12 [0.91, 1.37]	0.27	0.12
3 doses	2.45 [1.92, 3.13]	<0.0001	0.3	2.46 [1.93, 3.14]	<0.0001	0.30
Number of chronic Conditions *—*2 or more*	0.89 [0.80, 1]	0.043	0.05	-	-	-
Chronic conditions						
Allergies ^†^	-	-	-	0.89 [0.77, 1.02]	0.097	0.06
Diabetes	-	-	-	1.01 [0.85, 1.21]	0.87	0.09
Cardiovascular diseases	-	-	-	0.93 [0.82, 1.05]	0.24	0.06
Hypertension	-	-	-	0.87 [0.80, 0.95]	0.002	0.04
Respiratory diseases	-	-	-	1 [0.89, 1.11]	0.95	0.06
Diseases involving immune system	-	-	-	0.97 [0.86, 1.08]	0.55	0.06
Cancer	-	-	-	1.05 [0.87, 1.27]	0.60	0.10

* Reference categories: male (Gender), never infected (Previous COVID-19 infection), 1 dose (Number of vaccine doses), 0 or 1 (Number of chronic conditions). ^†^ Available in all countries but Italy.

## Data Availability

The availability of transfer of data will always be anonymous, upon request to the principal researcher of each cohort and based on the informed consent approved in each cohort and country.

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
