# Peer review of "Multimorbidity and Serological Response to SARS-CoV-2 Nine Months after 1st Vaccine Dose: European Cohort of Healthcare Workers—Orchestra Project"

_vaccines, 2023, doi:10.3390/vaccines11081340_

Round 1

Reviewer 1 Report

This report is useful considering the large number of vaccination performed during Corona Pandemic .More of such reports would help to understand how to control possible morbidities in future large scale vaccinations. However I  and perhaps others would like to see the brand of vaccines used in present study since it seems there should be a somehow relationship in between the vaccine type and morbidities due to cardiovascular diseases .If there is no relation, it should be clearly mentioned .If yes, why not mentioning while the author clearly write that the brand and type of vaccine used are known. That would be much important since there is pathway in choosing mRNA or other direction otherwise in case of future pandemics .
Let us also know the vaccine brands and the relation between their use and morbidity ??

Author Response

Dear Editor,

Dear Reviewers,

We would like to thank the reviewers for the encouraging comments and suggestions raised in order to improve our manuscript. We have tried to address all comments and clarify some sections. We have rewritten the Abstract and most of the Discussion following the comments. Please find below our responses to each comment (in blue) and the changes in the reviewed manuscript with track changes.

External Peer-Review Report(s)

Reviewer 1:
General comments
=============
This report is useful considering the large number of vaccination performed during Corona Pandemic .More of such reports would help to understand how to control possible morbidities in future large scale vaccinations. However I  and perhaps others would like to see the brand of vaccines used in present study since it seems there should be a somehow relationship in between the vaccine type and morbidities due to cardiovascular diseases .If there is no relation, it should be clearly mentioned .If yes, why not mentioning while the author clearly write that the brand and type of vaccine used are known. That would be much important since there is pathway in choosing mRNA or other direction otherwise in case of future pandemics .

Let us also know the vaccine brands and the relation between their use and morbidity ??

Response:  Thank you for your comment. We have added to Table 1 the vaccination scheme (mRNA-based vaccines, Adenovirus-based vaccines, or mixed type) per cohort. As Table 1 now shows, between 95% and 100% of individuals in each cohort, 98.4% of the total population considered, were vaccinated with mRNA-based vaccines. This coincides with the recommendation made by WHO for vaccination of healthcare workers1, which constitute the study population of this work. Although we agree with the reviewer that analysis would be of interest in the general population, when considering the small sample size of those vaccinated with adenovirus or mixed type vaccines in our study, performing any stratified analysis by vaccine type was not considered appropriate. 

References Rewiers answers

  1. WHO SAGE values framework for the allocation and priorization of COVID-19 vaccination. 14 September 2020 . Wold Health Organization.
  2. Collatuzzo, G.; Lodi, V.; Feola, D.; De Palma, G.; Sansone, E.; Sala, E.; Janke, C.; Castelletti, N.; Porru, S.; Spiteri, G.; et al. Determinants of Anti-S Immune Response at 9 Months after COVID-19 Vaccination in a Multicentric European Cohort of Healthcare Workers—ORCHESTRA Project. Viruses 2022, 14, 675. https://doi.org/10.3390/v14122657
  3. Collatuzzo G, Visci G, Violante FS, Porru S, Spiteri G, Monaco MGL, Larese Fillon F, Negro C, Janke C, Castelletti N, De Palma G, Sansone E, Mates D, Teodorescu S, Fabiánová E, Bérešová J, Vimercati L, Tafuri S, Abedini M, Ditano G, Asafo SS, Boffetta P; Orchestra WP5 Working Group. Determinants of anti-S immune response at 6 months after COVID-19 vaccination in a multicentric European cohort of healthcare workers - ORCHESTRA project. Front Immunol. 2022 Sep 29;13:986085. doi: 10.3389/fimmu.2022.986085.

Reviewer 2 Report

1. Abstract is not clear. This is not representing the main document. Please rewrite the abstract.

2. Please remove the bold for keywords.

3. Full form for SARS-CoV-2 was not updated in the introduction.

4. Discuss about the types of COVID-19 disease and write the prevalence of the disease in Italian population.

5. Please discuss the list of vaccines available globally.

6. Do authors have selected any 2 of the human diseases as multimorbidity or will they apt any 2 of the chronic diseases.

7. Introduction need to be describe in detail and connect the relation between multimorbidity and COVID-19.

8.  How does authors have selected the development of SARS-CoV-2 antibodies?

9. Is there any precise types for diabetes or mixed diabetes

10. Discussion was well-written.

Author Response

Dear Editor,

Dear Reviewers,

We would like to thank the reviewers for the encouraging comments and suggestions raised in order to improve our manuscript. We have tried to address all comments and clarify some sections. We have rewritten the Abstract and most of the Discussion following the comments. Please find below our responses to each comment (in blue) and the changes in the reviewed manuscript with track changes.

External Peer-Review Report(s)

Reviewer 2 :
General Comments
=================

  1. Abstract is not clear. This is not representing the main document. Please rewrite the abstract.

Response: We have now rewritten our abstract, hoping it is more representative now.

  1. Please remove the bold for keywords.

Response: We have removed the bold for keywords.

  1. Full form for SARS-CoV-2 was not updated in the introduction.

Response: Very appropriate observation, we have added the full name in the first line of the introduction section. Many thanks

  1. Discuss about the types of COVID-19 disease and write the prevalence of the disease in Italian population.

Response: Regarding this consideration, the ORCHESTRA study did not collect information on the severity of COVID-19 among HCWs. Consequently, we cannot compare this aspect, neither in the general Italian population nor with that of other countries.

  1. Please discuss the list of vaccines available globally.

Response: Based on your suggestion, we have included a reference in the second paragraph of the discussion section, pertaining to the vaccination guidelines recommendations for the autumn of 2023.

  1. Do authors have selected any 2 of the human diseases as multimorbidity or will they apt any 2 of the chronic diseases.

Response: All the diseases considered, which can be found listed in Table 1, were chronic conditions, as mentioned in the definition used for multimorbidity in the introduction section. Acute illnesses were discarded.

  1. Introduction need to be describe in detail and connect the relation between multimorbidity and COVID-19.

Response: We revised the introduction to meet the reviewer’s suggestion. Thank you.

  1. How does authors have selected the development of SARS-CoV-2 antibodies?

Response: Measurement methods for antibody levels differed among cohorts, but all of them were validated tools for the measurement of SARS-CoV-2 antibodies (see Table 2, where the assays are listed). We identified individuals with positive levels of SARS-CoV-2 antibodies according to the threshold defined by the assay’s manufacturer. It was not feasible to employ the same technique universally due to the unique circumstances in each country, and we have now mentioned it in the Methods section. However, using different assays between cohorts do not represent a limitation as the serological measurements were standardized across cohorts.

  1. Is there any precise types for diabetes or mixed diabetes

Response: The collection of comorbidities in most of the cohorts was through self-reported questionnaires, thus the diabetes type could vary among cohorts. We would rather consider diabetes as mixed diabetes. We have now addressed this aspect in the discussion section. Additionally, we have addressed this aspect in the discussion section.

  1. Discussion was well-written.

Response: Thank you for your comments and detailed review; it has increased the quality of our paper.

References Rewiers answers

  1. WHO SAGE values framework for the allocation and priorization of COVID-19 vaccination. 14 September 2020 . Wold Health Organization.
  2. Collatuzzo, G.; Lodi, V.; Feola, D.; De Palma, G.; Sansone, E.; Sala, E.; Janke, C.; Castelletti, N.; Porru, S.; Spiteri, G.; et al. Determinants of Anti-S Immune Response at 9 Months after COVID-19 Vaccination in a Multicentric European Cohort of Healthcare Workers—ORCHESTRA Project. Viruses 2022, 14, 675. https://doi.org/10.3390/v14122657
  3. Collatuzzo G, Visci G, Violante FS, Porru S, Spiteri G, Monaco MGL, Larese Fillon F, Negro C, Janke C, Castelletti N, De Palma G, Sansone E, Mates D, Teodorescu S, Fabiánová E, Bérešová J, Vimercati L, Tafuri S, Abedini M, Ditano G, Asafo SS, Boffetta P; Orchestra WP5 Working Group. Determinants of anti-S immune response at 6 months after COVID-19 vaccination in a multicentric European cohort of healthcare workers - ORCHESTRA project. Front Immunol. 2022 Sep 29;13:986085. doi: 10.3389/fimmu.2022.986085.

Reviewer 3 Report

The manuscript entitled "Multimorbidity and serological response to SARS-CoV-2 nine months after 1st vaccine dose. European cohort of healthcare workers – ORCHESTRA Project" is an interesting paper and brings new information regarding the vaccination. This topic is important and worth of publication.

The English quality if fine

Author Response

Dear Editor,

Dear Reviewers,

We would like to thank the reviewers for the encouraging comments and suggestions raised in order to improve our manuscript. We have tried to address all comments and clarify some sections. We have rewritten the Abstract and most of the Discussion following the comments. Please find below our responses to each comment (in blue) and the changes in the reviewed manuscript with track changes.

External Peer-Review Report(s)

Reviewer 3:
General comments
=============
The manuscript entitled "Multimorbidity and serological response to SARS-CoV-2 nine months after 1st vaccine dose. European cohort of healthcare workers – ORCHESTRA Project" is an interesting paper and brings new information regarding the vaccination. This topic is important and worth of publication.

Response: Thank you for your comments and encouraging comment.

References Rewiers answers

  1. WHO SAGE values framework for the allocation and priorization of COVID-19 vaccination. 14 September 2020 . Wold Health Organization.
  2. Collatuzzo, G.; Lodi, V.; Feola, D.; De Palma, G.; Sansone, E.; Sala, E.; Janke, C.; Castelletti, N.; Porru, S.; Spiteri, G.; et al. Determinants of Anti-S Immune Response at 9 Months after COVID-19 Vaccination in a Multicentric European Cohort of Healthcare Workers—ORCHESTRA Project. Viruses 2022, 14, 675. https://doi.org/10.3390/v14122657
  3. Collatuzzo G, Visci G, Violante FS, Porru S, Spiteri G, Monaco MGL, Larese Fillon F, Negro C, Janke C, Castelletti N, De Palma G, Sansone E, Mates D, Teodorescu S, Fabiánová E, Bérešová J, Vimercati L, Tafuri S, Abedini M, Ditano G, Asafo SS, Boffetta P; Orchestra WP5 Working Group. Determinants of anti-S immune response at 6 months after COVID-19 vaccination in a multicentric European cohort of healthcare workers - ORCHESTRA project. Front Immunol. 2022 Sep 29;13:986085. doi: 10.3389/fimmu.2022.986085.

Reviewer 4 Report

esults from the multi-centre Orchestra cohort have published before so investigators will be interested in the further measurements and analyses. As per the title it does contain data on immune responses in subjects that have more than one co-morbidity which is lacking in the published literature.

The regression analyses with single comorbidity roughly follow previous findings even though the the significance levels were only below 0.05 for hypertension and this was probably due to its much high prevalence in the study population than the other co-morbidities. The paper would be improved by explicitly discussing this point.

While the multimorbidity had lower titres the paper should also give the results of comparing versus people with single co-morbidities.

The allergy category needs a definition of what the criteria used for the group were. This is especially required to explain why the different centres had vastly different percentages of allergic subjects.

I cannot work out what the the first series (the different centres)  of relevant risks are in Table 3.  Why is the Munich 0.13 (and highly significant) while the other centre are positive? Why is there a line between  Slovakia and Spain?

Lines 295 -302 emphasise the importance of standardising the titres. While there would a tendency for dividing the mean with the standard error to do this the references given do not provide a mathematical explanation and there is no validation made with a standard serum that could be used for the different assay procedures. Also there are some very significant differences in the size of the titres given for different centres. Some comment on this is needed.

Author Response

Dear Editor,

Dear Reviewers,

We would like to thank the reviewers for the encouraging comments and suggestions raised in order to improve our manuscript. We have tried to address all comments and clarify some sections. We have rewritten the Abstract and most of the Discussion following the comments. Please find below our responses to each comment (in blue) and the changes in the reviewed manuscript with track changes.

Reviewer 4 :
General Comments
=================

Results from the multi-centre Orchestra cohort have published before so investigators will be interested in the further measurements and analyses. As per the title it does contain data on immune responses in subjects that have more than one co-morbidity which is lacking in the published literature. 

Response: Thank you

The regression analyses with single comorbidity roughly follow previous findings even though the the significance levels were only below 0.05 for hypertension and this was probably due to its much high prevalence in the study population than the other co-morbidities. The paper would be improved by explicitly discussing this point.

Response: We agree with the reviewer’s comment, and we have now commented it in the discussion. We would like to highlight that the study population consist of healthcare workers, which might be healthier than the general population. Therefore, the prevalence of chronic conditions might be lower than that of the general population.

While the multimorbidity had lower titres the paper should also give the results of comparing versus people with single co-morbidities.

Response:  In this work we seek to study the effect of multimorbidity on the levels of IgG-spike antibodies 9 months after vaccination. The definition for multimorbidity used in this study is that of having 2 or more concurrent chronic diseases. Therefore, although we agree with the reviewer that it would be very interesting to study the effect of each of the chronic conditions on their own, it is out of the scope of this work.

The allergy category needs a definition of what the criteria used for the group were. This is especially required to explain why the different centres had vastly different percentages of allergic subjects. 

Response: The presence of allergy was obtained through self-reported records in most cohorts and electronic health records, but its definition was not consistent across cohorts. In the methods section, we have included details on how the multimorbidity information is collected. Additionally, we have addressed this aspect in the discussion section.

I cannot work out what the the first series (the different centres)  of relevant risks are in Table 3.  Why is the Munich 0.13 (and highly significant) while the other centre are positive? Why is there a line between  Slovakia and Spain?

Response: The line between Slovakia and Spain was a mistake that has been corrected in the new version. Thank you for noting it.

Regarding the differences between centres, we would like to refer to our response to the next reviewer’s comment. These differences are related to the variability in the standardized means for each centre. Most of them are quite homogeneous, except for Munich and Padova. This difference of Munich has already been reported in previous reports from the ORCHESTRA cohort2. Slovakia, on the other hand, might show a high variability due to the low sample size.

Lines 295 -302 emphasise the importance of standardising the titres. While there would a tendency for dividing the mean with the standard error to do this the references given do not provide a mathematical explanation and there is no validation made with a standard serum that could be used for the different assay procedures. Also there are some very significant differences in the size of the titres given for different centres. Some comment on this is needed.

Response: Regarding the heterogeneity in blood tests among different centers, we would like to note that it is a common issue in studies on COVID-19, given the adoption of different detection methods in different countries and even hospitals of the same country. We understand the reviewer’s concern but, to the best of our knowledge, no specific approach to overcome this problem has been proposed. The one we used partially addresses the problem by making the antibody levels comparable. In this way we were able to identify actual differences in serological response among the included cohorts according to various characteristics. As stated in previous papers from the ORCHESTRA cohort, this method could be useful also for other analyses where the data have been collected in different ways to make measurements comparable2,3,.

References Rewiers answers

  1. WHO SAGE values framework for the allocation and priorization of COVID-19 vaccination. 14 September 2020 . Wold Health Organization.
  2. Collatuzzo, G.; Lodi, V.; Feola, D.; De Palma, G.; Sansone, E.; Sala, E.; Janke, C.; Castelletti, N.; Porru, S.; Spiteri, G.; et al. Determinants of Anti-S Immune Response at 9 Months after COVID-19 Vaccination in a Multicentric European Cohort of Healthcare Workers—ORCHESTRA Project. Viruses 2022, 14, 675. https://doi.org/10.3390/v14122657
  3. Collatuzzo G, Visci G, Violante FS, Porru S, Spiteri G, Monaco MGL, Larese Fillon F, Negro C, Janke C, Castelletti N, De Palma G, Sansone E, Mates D, Teodorescu S, Fabiánová E, Bérešová J, Vimercati L, Tafuri S, Abedini M, Ditano G, Asafo SS, Boffetta P; Orchestra WP5 Working Group. Determinants of anti-S immune response at 6 months after COVID-19 vaccination in a multicentric European cohort of healthcare workers - ORCHESTRA project. Front Immunol. 2022 Sep 29;13:986085. doi: 10.3389/fimmu.2022.986085.